# FIRST-line support for assistance in breathing in children (FIRST-ABC): a master protocol of two randomised trials to evaluate the non-inferiority of high-flow nasal cannula (HFNC) versus continuous positive airway pressure (CPAP) for non-invasive respiratory support in paediatric critical care

Alvin Richards-Belle [1], Peter Davis,[2] Laura Drikite,[1] Richard Feltbower,[3] Richard Grieve,[4] David A Harrison,[1] Julie Lester,[5] Kevin P Morris,[6] Paul R Mouncey,[1] Mark J Peters,[7,8] Kathryn M Rowan,[1] Zia Sadique,[4] Lyvonne N Tume,[9] Padmanabhan Ramnarayan [10]

For numbered affiliations see end of article.

**Correspondence to**
Dr Padmanabhan Ramnarayan;
p.ramnarayan@gosh.nhs.uk

## ABSTRACT

**Introduction** Even though respiratory support is a common intervention in paediatric critical care, there is no randomised controlled trial (RCT) evidence regarding the effectiveness of two commonly used modes of non-invasive respiratory support (NRS), continuous positive airway pressure (CPAP) and high-flow nasal cannula therapy (HFNC). FIRST-line support for assistance in breathing in children is a master protocol of two pragmatic non-inferiority RCTs to evaluate the clinical and cost-effectiveness of HFNC (compared with CPAP) as the first-line mode of support in critically ill children.

**Methods and analysis** We will recruit participants over a 30-month period at 25 UK paediatric critical care units (paediatric intensive care units/high-dependency units). Patients are eligible if admitted/accepted for admission, aged >36 weeks corrected gestational age and <16 years, and assessed by the treating clinician to require NRS for an acute illness (step-up RCT) or within 72 hours of extubation following a period of invasive ventilation (step-down RCT). Due to the emergency nature of the treatment, written informed consent will be deferred to after randomisation. Randomisation will occur 1:1 to CPAP or HFNC, stratified by site and age (<12 vs ≥12 months). The primary outcome is time to liberation from respiratory support for a continuous period of 48 hours. A total sample size of 600 patients in each RCT will provide 90% power with a type I error rate of 2.5% (one sided) to exclude the prespecified non-inferiority margin of HR of 0.75. Primary analyses will be undertaken separately in each RCT in both the intention-to-treat and per-protocol populations.

**Ethics and dissemination** This master protocol received favourable ethical opinion from National Health Service East of England—Cambridge South Research Ethics Committee (reference: 19/EE/0185) and approval from the Health

## Strengths and limitations of this study

▶ FIRST-line support for assistance in breathing in children (FIRST-ABC) is a master protocol of the two largest randomised controlled trials (RCTs) to date to study the clinical and cost-effectiveness of high-flow nasal cannula as the first-line mode of non-invasive respiratory support in critically ill children.

▶ The FIRST-ABC master protocol includes two separate RCTs, one in acutely ill children requiring respiratory support (step-up RCT) and one in children requiring respiratory support after extubation from invasive ventilation (step-down RCT), to address the research question in two distinct but common clinical scenarios.

▶ The design and conduct of FIRST-ABC has been informed by a successful pilot RCT that confirmed the feasibility of delivering a large pragmatic trial in critically ill children.

▶ The choice of the primary outcome, time to liberation from all forms of respiratory support for a continuous period of at least 48 hours, was informed by clinicians as well as through patient and public involvement.

▶ Changes to clinical practice during the trial period, and a resultant shift in equipoise regarding the choice of first-line mode of respiratory support in critically ill children, may affect the ability to recruit successfully to the RCTs.

Research Authority (reference: 260536). Results will be disseminated via publications in peer-reviewed medical journals and presentations at national and international conferences.

**Trial registration number** ISRCTN60048867

## INTRODUCTION

Nearly 75% of the 20 000 critically ill children admitted annually to UK paediatric intensive care units (PICUs) receive some form of respiratory support.[1] Increasing recognition of the risks of invasive ventilation has prompted greater use of non-invasive respiratory support (NRS) worldwide.[1 2] Two main modes of NRS are used, to support acutely ill children with respiratory failure or to provide postextubation support after a spell of invasive ventilation.

Continuous positive airway pressure (CPAP) has been used by PICUs for over three decades.[3–5] Although observational data suggest that CPAP is effective, there have been few randomised controlled trials (RCTs) of CPAP in critically ill children.[5–7] CPAP can be uncomfortable and may be associated with complications such as air-leak and nasal trauma, often necessitating the use of sedation, close monitoring and a high level of nursing input. An alternate mode of NRS, high-flow nasal cannula (HFNC), has gained popularity more recently. It appears easy to use and is well tolerated.[8–11] Between 16% and 35% of PICU admissions receive HFNC at some point during their stay.[1 12 13] The potential benefits of HFNC (improved patient comfort, safety profile and ease of nursing care) must be balanced against its potential risks (air leak, abdominal distension and nosocomial infection), and concerns regarding unnecessary prolongation of PICU/hospital stay and excess mortality from delayed escalation. There are few RCTs comparing HFNC with CPAP in the PICU setting. Previous RCTs do not include children with a range of ages and diagnoses needing either step-up or step-down (postextubation) care, making it impossible to generalise their findings to contemporary practice.[14–16]

FIRST-line support for assistance in breathing in children (FIRST-ABC), therefore, addresses an important clinical dilemma faced daily by critical care clinicians: in a child requiring NRS, which modality, HFNC or CPAP, should they use as first-line therapy to achieve the best patient outcomes? Our research question was prioritised by clinicians as well as parents/patients. We previously successfully completed a pilot RCT, which supported the feasibility of performing a large pragmatic RCT comparing CPAP and HFNC in critically ill children, and informed its design and conduct.[17] This protocol has been written in accordance with the Standard Protocol Items: Recommendations for Interventional Trials statement.[18]

## METHODS

### Hypothesis

In critically ill children assessed by the treating clinician to require NRS, first-line use of HFNC is non-inferior to CPAP in time to liberation from respiratory support.

### Aim

To evaluate the clinical and cost-effectiveness of the use of HFNC, as compared with CPAP, when used as the first-line mode in critically ill children requiring NRS:

1. For an acute illness (step-up RCT).
2. Within 72 hours of extubation following a period of invasive ventilation (step-down RCT).

### Primary objective

To evaluate the non-inferiority of HFNC, as compared with CPAP, when used as the first-line mode of NRS, both as a step-up treatment (step-up RCT) and as a step-down treatment (step-down RCT), on the time to liberation from respiratory support.

### Design

FIRST-ABC is a master protocol comprising two pragmatic, multicentre, parallel groups, non-inferiority RCTs (step-up RCT and step-down RCT) with shared infrastructure, including an internal pilot stage and integrated health economic evaluation. This design allows the research question to be addressed in each of the two important populations in an efficient way by minimising time and infrastructure costs as compared with conducting two sequential RCTs.[19] The pragmatic study design ensures that research findings can be more easily generalised to real-world practice.

A non-inferiority design was chosen based on previous RCTs in this area and feedback from clinicians from the UK Paediatric Intensive Care Society—Study Group in July 2017 which indicated that the potential benefits of HFNC (in terms of patient comfort and ease of use) would mean that it would likely be preferred in usual practice even if not shown to be superior to CPAP.

### Setting

FIRST-ABC is set in National Health Service (NHS) paediatric critical care units (PICU and/or high-dependency units (HDUs)) across England, Wales and Scotland. General medical-surgical, cardiac and mixed units were considered for participation. Sites are eligible to take part if they confirm collective equipoise regarding the choice of first-line NRS in their unit and commit to following trial procedures, including randomisation and data collection. Sites can start recruitment only after a site initiation visit and all relevant regulatory approvals.

### Population

Critically ill children assessed by the treating clinician to require NRS for (A) an acute illness (step-up RCT) or (B) within 72 hours of extubation following a period of invasive ventilation (step-down RCT).

### Screening

Potentially eligible patients admitted/accepted for admission to the participating critical care unit will be screened against the inclusion/exclusion criteria by the local clinical/research team. For the step-up RCT, all admissions to the unit will be screened. For the step-down

RCT, all patients extubated during unit admission will be screened. From these, Screening and Enrolment Logs will record enrolled patients, reasons for exclusion and reasons eligible patients are not enrolled.

### Inclusion criteria
1. Admitted/accepted for admission to PICU/HDU.
2. Age>36 weeks corrected gestational age and <16 years.
3. Assessed by the treating clinician to require NRS.
    1. For an acute illness (step-up RCT).
    2. Within 72 hours of extubation following a period of invasive ventilation (step-down RCT).

### Exclusion criteria
1. Assessed by the treating clinician to require immediate intubation and invasive ventilation due to severe hypoxia, acidosis and/or respiratory distress, upper airway obstruction, inability to manage airway secretions or recurrent apnoeas.
2. Tracheostomy in place.
3. Received HFNC/CPAP for >2 hours in the prior 24 hours.
4. On home non-invasive ventilation prior to PICU/HDU admission.
5. Presence of untreated air-leak (pneumothorax/pneumomediastinum).
6. Midfacial/craniofacial anomalies (unrepaired cleft palate, choanal atresia) or recent craniofacial surgery.
7. Agreed 'not for intubation' or other limitation of critical care treatment plan in place.
8. Previously recruited to FIRST-ABC (step-up RCT or step-down RCT on this or a previous admission).
9. Clinician decision to start other form of NRS (ie, not HFNC or CPAP, eg, bilevel positive pressure and negative pressure ventilation).

### Randomisation
Randomisation will be performed after confirming eligibility and as close as possible to the anticipated start of the randomised treatment. In each RCT, eligible patients will be randomised in a 1:1 ratio to either CPAP or HFNC using a central telephone/web-based randomisation service available 24 hours/7 days a week. The randomisation sequence will be computer generated and variable block sizes will be used to strengthen allocation concealment. Randomisation will be stratified by site and age (<12 months vs ≥12 months) to minimise imbalance arising from unit practices and interface selection.

The randomised treatment will be commenced as soon as practically possible. Following randomisation, each participant will be assigned a unique FIRST-ABC trial number and a case report form (CRF) completed by the local research team.

### Delivery of HFNC
Any approved medical device capable of delivering heated, humidified, high flow through nasal cannulae can be used to provide HFNC at the prescribed gas flow rates during the trial period. To standardise treatment, clinical criteria and guidance for the initiation, maintenance and weaning of HFNC are provided in a trial algorithm (figure 1). The trial algorithms were developed iteratively in consultation with paediatric critical care clinicians across the UK (both via email and in person at a Collaborators' Meeting held prior to the start of the trial).

The trial recommends that patients are assessed for response to the treatment, readiness to wean and for stopping HFNC, as per the HFNC algorithm, at least twice per day (eg, at ward rounds).

### Delivery of CPAP
CPAP will be started using an approved medical device at a set expiratory pressure of 7–8 cm $H_2O$. The trial does not specify any particular device or patient interface for the provision of CPAP. To standardise treatment, clinical criteria and guidance for the initiation, maintenance and weaning of CPAP are provided in a trial algorithm (figure 2). It is recommended that patients are assessed for response to the treatment, readiness to wean and for stopping CPAP, as per the CPAP algorithm, at least twice per day (eg, at ward rounds).

### Clinical practice during the trial
Since staff in participating sites already use HFNC and CPAP, no additional central training related to the use of HFNC or CPAP will be provided for the trial, but resources for training in the trial algorithms will be provided. As the medical devices and interfaces that deliver HFNC and CPAP are easily distinguishable from each other, it will not be possible to blind the patient, parents/guardians or clinical staff.

The trial algorithms will be followed until the patient has been liberated from all forms of respiratory support for at least 48 continuous hours. As per current practice, clinicians will be able to stop HFNC/CPAP and switch to the other treatment or escalate to other forms of respiratory support, if clinically deemed necessary. Prespecified objective criteria to identify non-responders to HFNC/CPAP are provided in the algorithms as a guide for clinicians considering switching or escalating respiratory support. Reasons for switches or escalations will be recorded. Patients who switch or escalate treatments will remain in the trial and continue to be monitored until liberation from respiratory support. All other usual care (eg, sedation, feeding) will be at the discretion of the treating clinical team.

### Consent procedures
Consent will be sought for the child (patient) from their parent/legal guardian. Children become eligible for FIRST-ABC when critically ill, a profoundly stressful time for parents/guardians, during which there are ethical concerns both about the burden of trying to understand the trial and the ability to provide informed consent. Initiation of NRS typically occurs during a time-sensitive

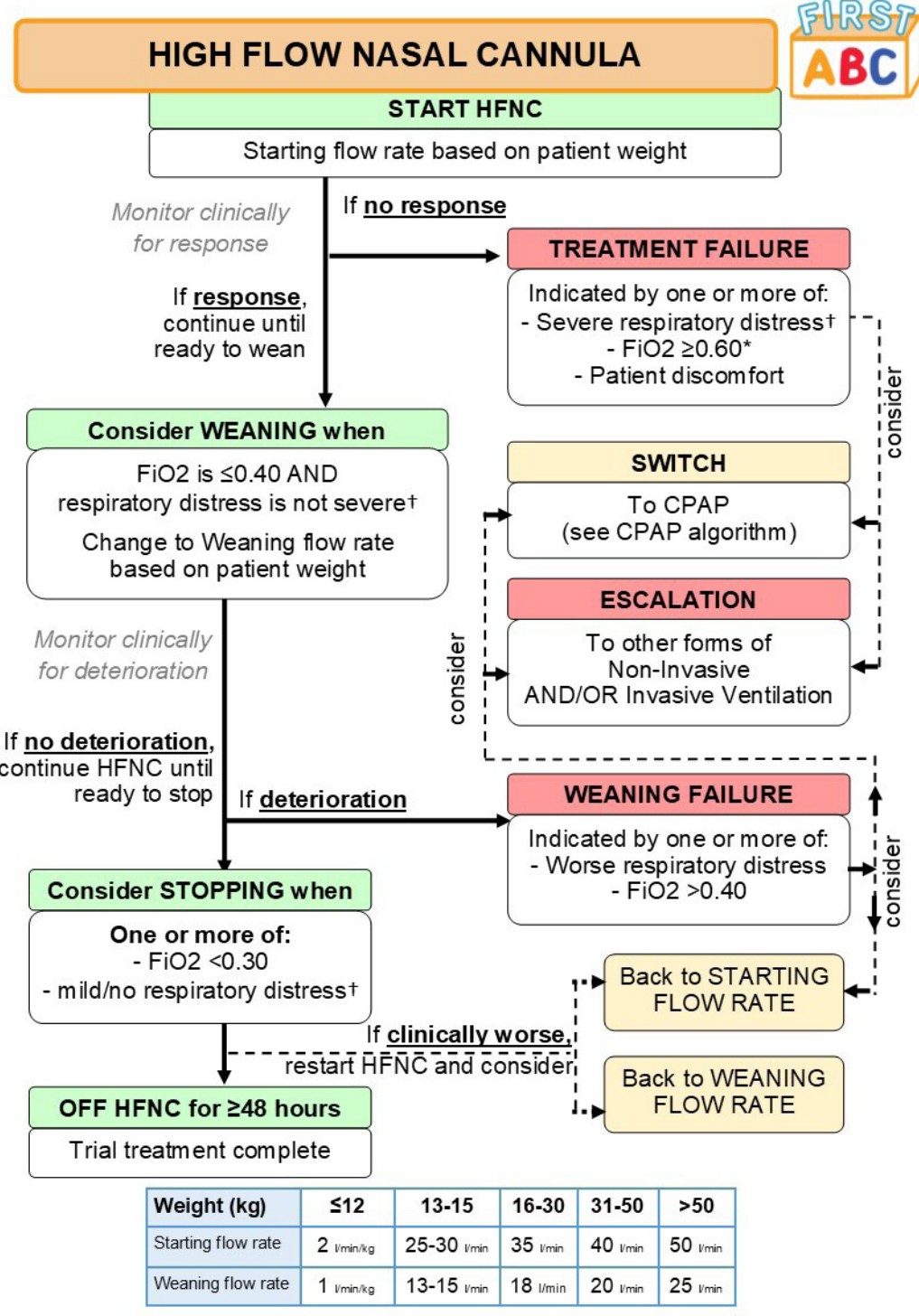

**Figure 1** Algorithm for delivery of HFNC. *Titrate $FiO_2$ while on HFNC to maintain peripheral oxygen saturations ($SpO_2$) ⩾92% (or patient-specific target). †Respiratory distress defined as: mild (one accessory muscle used, mild indrawing of subcostal and intercostal muscles, mild tachypnoea, no grunting), moderate (two accessory muscles used, moderate indrawing of subcostal and intercostal muscles, moderate tachypnoea, occasional grunting); or severe (use of all accessory muscles, severe indrawing of subcostal and intercostal muscles, severe tachypnoea, regular grunting). CPAP, continuous positive airway pressure; FIRST-ABC, FIRST-line support for assistance in breathing in children; $FiO_2$, fractional inspired oxygen; HFNC, high-flow nasal cannula.

situation, where delays could be detrimental to the child and to the trial's scientific validity. Moreover, both CPAP and HFNC are already widely used in standard practice across the NHS. Considering these reasons, FIRST-ABC has been given ethical approval to use a deferred consent model ('research without prior consent'). Once a patient is confirmed eligible, they will be randomised and the allocated treatment (CPAP or HFNC) commenced as

**Figure 2** Algorithm for delivery of continuous positive airway pressure (CPAP). *Titrate $FiO_2$ while on HFNC to maintain $SpO_2 \geqslant 92\%$ (or patient-specific target). †Respiratory distress defined as: mild (one accessory muscle used, mild indrawing of subcostal and intercostal muscles, mild tachypnoea, no grunting), moderate (two accessory muscles used, moderate indrawing of subcostal and intercostal muscles, moderate tachypnoea, occasional grunting); or severe (use of all accessory muscles, severe indrawing of subcostal and intercostal muscles, severe tachypnoea, regular grunting). FAST-ABC, FIRST-line support for assistance in breathing in children; $FiO_2$, fractional inspired oxygen; HFNC, high-flow nasal cannula.

soon as possible. This model, developed in line with the CONSeNt methods in paediatric Emergency and urgent Care Trials guidance,[20] has been found acceptable to parents/guardians and clinicians in several recent RCTs in the PICU setting[17 21–25] and is informed by experience/feedback from the pilot RCT.[17]

Following randomisation, a trained, delegated member of the local research team will approach the child's parents/guardians as soon as appropriate and practically possible to discuss the trial (usually within 24–48 hours of randomisation). A participant information sheet will be provided, covering information about the purpose of the trial; the consequences of participating or not;

confidentiality; use of personal data; data security and the future availability of the trial results. A consent form (see online supplementary file 1) will be provided, indicating that: the information given has been read and understood; participation is voluntary and consent can be withdrawn at any time without consequence; and that consent is given for access to medical records to continue data collection, to receive a follow-up questionnaire and for anonymised data to be shared in the future. Due to age and severity of illness, it will not be possible to involve the patient in the consenting process. Instead, assent will be obtained prior to hospital discharge if their condition allows (eg, they regain mental capacity).

A modification of the consent procedure will be utilised for two rare situations where either the patient: (1) is discharged from hospital prior to obtaining consent or (2) dies prior to consent being sought.[24 26] In the former, the local research team will follow up with the parent/guardian, initially by phone and then by post, for consent. Postal contact will be made again if there is no response after 4 weeks. If no consent form is received within 4 weeks of the second letter, the participant will be included in the trial unless they notify the research team otherwise. In the latter situation, the local research team will obtain information from colleagues and bereavement counsellors to establish the most appropriate clinical/research team member to notify the parents/guardians of involvement in the trial. If approach for consent is deemed not appropriate prior to the parent/guardian's departure from hospital, then they will be approached by post 4 weeks postrandomisation. The letter will explain how to opt out of the trial. Postal contact will be made again if there is no response after 4 weeks. If no consent form is received within 4 weeks of the second letter, the participant's data will be included in the trial.

If informed consent is refused or withdrawn, this decision will be respected and abided by, and no further contact made. All data occurring up to the point of this decision will be retained in the trial, unless parents/guardians request otherwise.

### Safety monitoring

Adverse event (AE) reporting will follow the Health Research Authority (HRA) guidelines on safety reporting in studies which do not useClinical Trials of Investigational Medicinal Products. The following events have been prespecified as potential AEs that could be related to CPAP and/or HFNC and observed in participants from the date and time of randomisation until 48 hours of liberation from all forms of respiratory support:

1. Nasal trauma.
2. Facial/neck trauma.
3. Abdominal distension.
4. Pneumothorax.
5. Pneumomediastinum.
6. Subcutaneous emphysema.
7. Facial thermal injury.
8. Respiratory arrest.

9. Cardiac arrest.
10. Aspiration

Occurrences of the specified, expected AEs will be recorded for all randomised patients. Considering that eligible patients are critically ill and at increased risk of experiencing AEs, occurrences of non-specified, AEs will only be reported if considered to be related to either CPAP or HFNC (ie, 'possibly', 'probably' or 'definitely' related). Any event classified as 'severe' or 'life-threatening' in severity is considered a serious AE (SAE) and must be reported to Intensive Care National Audit and Research Centre (ICNARC) Clinical Trials Unit (CTU). If the SAE is evaluated by the trial management group (TMG) as a related and unexpected SAE, the ICNARC CTU will submit a report to the Research Ethics Committee (REC) within 15 calendar days.

### Questionnaire follow-up

At 6 months, after assessing the child's survival status, each consenting parent will be sent a questionnaire (via email or post) by the ICNARC CTU to assess health-related quality of life (HrQoL) and health service/resource use. Non-responders will be followed up by telephone 3 weeks later.

## OUTCOME MEASURES
### Primary outcome

Time to liberation from respiratory support, defined as the start of a 48-hour period during which the child was free of all forms of respiratory support.

The primary outcome definition of respiratory support does not include administration of supplementary oxygen alone. In addition, the primary outcome will to be monitored/recorded after discharge from critical care, as necessary. We chose time to liberation from respiratory support, instead of rate of (re)intubation, as the primary outcome for several reasons, including: (1) through our patient and public involvement (PPI) work, parents/families reported that even though intubation was clearly an undesirable outcome, the fact that the child needed a 'breathing machine' of any description would be more important for them, in terms of assessing the success or failure of the intervention. Normalisation of 'breathing' was an important outcome prioritised over intubation; (2) since the rate of intubation on average was around 20% in the pilot RCT, nearly 80% of patients may not fulfil the intubation outcome. In these patients, several non-invasive support modes may be used, which prolong the time the patient is on 'breathing support' with resource implications for critical care. Clinicians felt that it was important that the effect of the intervention was assessed on patients who did not need intubation as well as on those who did. (3) unpublished data from the pilot RCT showed that the length of respiratory support is longer in patients who need intubation compared with

those who do not. Therefore, the adverse impact of intubation is likely reflected in longer duration of respiratory support.

### Secondary outcomes

► Mortality at PICU/HDU discharge, day 60 and day 180.
► Rate of (re) intubation at 48 hours.
► Duration of PICU/HDU and hospital stay.
► Patient comfort, during randomised treatment and during NRS (ie, HFNC and/or CPAP), assessed using the COMFORT-B score.[27]
► Proportion of patients in whom sedation is used during NRS.
► Parental stress, in hospital at/around the time of consent at 24–48 hours, measured using the Parental Stressor Scale: PICU.[28]
► HrQoL at 6 months using age-appropriate Paediatric Quality of Life Inventory (PedsQL)[29] and Child Health Utility (CHU-9D) questionnaire.[30]

### Cost-effectiveness analysis outcomes

► Total costs at 6 months.
► Quality-adjusted life-years (QALYs) at 6 months.
► Incremental net monetary benefit gained at a willingness-to-pay of £20 000 per QALY at 6 months associated with HFNC versus CPAP.[31]

### Data collection

To maximise efficiency, FIRST-ABC collaborates with the Paediatric Intensive Care Audit Network (PICANet) to make best use of established PICU data collection infrastructure. Where possible, recruited patients will be consented for data linkage with routine sources (eg, national death registration data via NHS Digital or equivalent). Additional trial-specific data collection items are limited to the minimum required to deliver trial objectives (table 1).

All participant data will be entered onto the secure electronic CRF and undergo validation checks for completeness, accuracy and consistency. The site principal investigator will oversee and be responsible for data collection, quality and recording.

## STATISTICAL METHODS

### Sample size

To achieve 90% power with a type I error rate of 2.5% (one sided) to exclude the pre-specified non-inferiority margin of HR=0.75 requires 508 events to be observed. Based on pilot RCT data,[17] we anticipate 5% censoring due to death or transfer, leading to a required sample size of 268 patients per group in each of the two RCTs. To allow for withdrawal/refusal of consent, and for exclusion due to non-adherence in the per-protocol (PP) population, we will recruit a total sample size of 600 patients in each RCT.

### Internal pilot

Data will be analysed at the end of the internal pilot stage (months 7–12 of the grant timeline) on patients recruited during the first 6 months in each RCT. The RCTs will progress from pilot to full trial based on prespecified progression criteria related to successful site set-up, screening and recruitment, and adherence. The final decision on progression will be made by the funder after

| Table 1 | Patient data collection schedule | | | | | |
|---|---|---|---|---|---|---|
| | **Baseline** | **At time of consent** | **During non-invasive respiratory support** | **End of PICU/ HDU stay** | **End of hospital stay** | **At 6 months** |
| In-hospital | | | | | | |
| Clinical/baseline data | ✔ | | | | | |
| Patient/parent details | | ✔ | | | | |
| Types of respiratory support received* | ✔ | | ✔ | | | |
| Patient comfort and sedation use | | | ✔ | | | |
| Parental stress | | ✔ | | | | |
| Discharge data | | | | ✔ | ✔ | |
| Safety monitoring data | | | ✔ | | | |
| At follow-up | | | | | | |
| PedsQL | | | | | | ✔ |
| CHU-9D | | | | | | ✔ |
| Health services/resource use | | | | | | ✔ |

*Including weaning, switches and escalations from high flow nasal cannula/continuous positive airway pressure.
CHU-9D, Child Health Utility-9D; HDU, high-dependency unit; PedsQL, Paediatric Quality of Life Inventory; PICU, paediatric intensive care unit.

recommendation, or not, by the trial steering committee (TSC).

## Clinical effectiveness analysis

All analyses will be publicly lodged[32] in a statistical analysis plan, a priori, before the investigators are unblinded to any trial outcomes. Following best practice for non-inferiority trials, the primary analyses will be undertaken in both intention-to-treat (ITT) and PP populations, with robust conclusions possible in the situation where both populations provide concordant results. Results will be reported in accordance with the Consolidated Standards of Reporting Trials statement extensions for non-inferiority and pragmatic trials.[33 34]

Analyses will be undertaken independently for each RCT. In each RCT, baseline patient characteristics will be compared between the two groups to observe balance and the success of randomisation. These comparisons will not be subjected to statistical testing. The delivery of the intervention will be described for each group in detail, including (but not limited to) number and percentage of patients who commence the randomised treatment, remain on the randomised treatment until liberation from ventilation, who are changed to a different mode of respiratory support.

HFNC will be considered non-inferior to CPAP if the lower bounds of the 95% CIs for the HR from Cox regression models on time to liberation from respiratory support fitted in both the ITT and PP populations exclude the prespecified non-inferiority margin of 0.75 (corresponding to approximately a 16-hour increase in median time to liberation, based on pilot RCT data). This margin was considered adequate such that the other potential benefits of HFNC in terms of comfort and tolerability would mean that it would be likely to be preferred in usual practice. The Cox regression models will be adjusted for important baseline characteristics. The covariates for inclusion in the regression models will be selected a priori based on an established relationship with outcome for critically ill children, and not because of observed imbalance, significance in univariable analyses or by a stepwise selection method.

Subgroup analyses for the primary outcome will be performed to test for interactions between the effect of allocated treatment group and the following baseline covariates:
► Age (<12 months vs ≥12 months).
► Severity of respiratory distress at randomisation (severe vs mild/moderate).
► Comorbidities (none vs neurological/neuromuscular vs other).
► Sp02/fractional inspired oxygen (SF) ratio at randomisation.
► For step-up RCT only:
  – Clinical indication (bronchiolitis vs other respiratory (airway problem, asthma/wheeze or any other respiratory) versus cardiac versus other (neurological, sepsis/infection, any other)).

  – Whether child was on NRS at randomisation (yes/no).
► For step-down RCT only:
  – Length of prior invasive mechanical ventilation (<5 days vs ≥5 days).
  – Reason for invasive mechanical ventilation (cardiac vs other).
  – Planned (randomisation followed by extubation) versus rescue (extubation followed by randomisation) NRS.

We will treat age as a continuous variable and determine whether the model goodness-of-fit is better versus treating age as a categorical term for any analyses focusing on those over the age of 12 months. We anticipate a high proportion of patients will be aged <12 months, and therefore, exploration of age effects in the older ages will only be conducted if there are sufficient patient numbers.

As a sensitivity analysis, the primary analysis will be repeated using time to start weaning of NRS (ie, duration of 'acute' respiratory support) and time to meeting objective 'readiness to wean NRS' criteria.

Secondary analyses of binary outcomes (mortality, reintubation) will be performed by Fisher's exact test and adjusted logistic regression. Duration of survival to day 180 will be plotted as Kaplan-Meier survival curves, compared unadjusted with the log rank test and adjusted using Cox regression models. Analyses of duration of PICU/HDU and hospital stay will be performed by Wilcoxon rank-sum tests, stratified by survival status. Analyses of COMFORT-B score, sedation use, PSS:PICU and HrQoL will be performed by t-tests and adjusted linear regression.

In the step-up RCT, a single interim analysis will be carried out after the recruitment and follow-up to day 60 of 300 patients. The interim analysis will use a Peto-Haybittle stopping rule to recommend early termination due to superiority of either intervention (p<0.001) in time to liberation from respiratory support or evidence of harm from either intervention (p<0.05) in mortality at day 60. Both tests will be performed using a log-rank test on all available data within the ITT population. Further interim analyses will be performed only if requested by the data monitoring and ethics committee (DMEC).

In the step-down RCT, due to faster than anticipated recruitment, no formal interim analysis will be performed. Safety data (counts and percentages of AEs by arm and a line listing of SAEs) will be available for scrutiny by the DMEC, by the end of the internal pilot stage.

## Integrated health economic evaluation

The cost-effectiveness analysis (CEA) will take an NHS and Personal Social Services perspective.[31] Patient-level resource use data will be obtained from CRFs, PICANet and a parent-completed Health Services Questionnaire (HSQ). Resource use data from the PICU/HDU stay will be taken from the CRF and linked routine data

from PICANet. Information on subsequent PICU/HDU and hospital admissions will be obtained via data linkage with PICANet and NHS Digital and also through completion of the HSQ. Data on the level of care for PICU bed-days will be gathered through routine collection of the Paediatric Critical Care Minimum Dataset in the participating sites via the PICANet database. Use of primary care and community health services will be assessed by HSQ at 6 months. Patient-level resource use data will be combined with appropriate unit costs from the NHS payment by results and Personal Social Services Research Unit databases to report total costs per patient for up to 6 months postrandomisation. Data from PedsQL and CHU-9D at 6 months will be combined with survival data to report QALYs at 6 months. The CEA will follow the ITT principle and report the mean (95% CI) incremental costs, QALYs and net monetary benefit at 6 months. The CEA will use multilevel linear regression models that allow for clustering of patients at site. The analysis will adjust for key baseline covariates at both patient and site level.

## ETHICS AND DISSEMINATION
### Research ethics
The trial received favourable ethical opinion from NHS East of England—Cambridge South Research Ethics Committee (reference number: 19/EE/0185) and approval from the HRA (Integrated Research Application System (IRAS) number: 260536). Evidence of local confirmation of capacity and capability at each site must be provided to the ICNARC CTU prior to site activation.

### Confidentiality
ICNARC CTU will act to preserve participant confidentiality and will not disclose or reproduce any information by which participants could be identified. All data will be stored securely.

### Oversight
The TMG, led by the chief investigator, is responsible for the management of FIRST-ABC. It meets regularly and includes the Investigators and ICNARC CTU trial team. FIRST-ABC is managed by the ICNARC CTU in accordance with the Medical Research Council's Good Research Practice: Principles and Guidelines[35] which is based on the International Conference on Harmonisation guidelines on Good Clinical Practice[36] principles and the UK Department of Health's Policy Framework for Health and Social Care Research.[37] The on-site monitoring plan will follow a risk-based strategy.

A majority independent TSC has been established to monitor trial progress and includes PPI representatives, experienced clinicians and researchers/statisticians, in addition to the chief investigator and head of research at ICNARC. An independent DMEC, comprising experienced clinicians and statisticians, has been established to monitor patient recruitment and retention, adherence and safety.

The Great Ormond Street Hospital for Children NHS Foundation Trust is the trial sponsor (reference: 17IA05). As the sponsor is an NHS organisation, NHS indemnity will apply for legal liability arising from the design, management and conduct of the research.

### Patient and public involvement
We had considerable PPI input into the pilot RCT[17] as well as the main trial described here. Following the pilot RCT, the PPI Group for Research at Great Ormond Street Hospital was consulted on the choice of the primary outcome for the main RCTs (see Outcome measures section). The parent of a child who received respiratory support is a coinvestigator and has actively contributed to the trial design and procedures, including the use of deferred consent and patient/parent information sheets and other materials.

### Trial status
This paper presents the master protocol (V.1.2, dated 23 January 2020)[38] for the two largest RCTs studying the clinical and cost effectiveness of HFNC therapy as the first-line mode of NRS in critically ill children. It will provide robust evidence for the two distinct but common clinical scenarios in which NRS is primarily used. The first participant was recruited in August 2019. At the time of submission, patient recruitment was ongoing—with recruitment planned to complete in November 2020 and January 2022 for the step-down RCT and step-up RCT, respectively. Each RCT will be disseminated independently, including through publication in peer-reviewed medical journals and at national and international conferences.

**Author affiliations**
[1]Clinical Trials Unit, Intensive Care National Audit and Research Centre, London, UK
[2]Paediatric Intensive Care, Bristol Royal Hospital for Children, Bristol, UK
[3]Leeds Institute for Data Analytics, University of Leeds, Leeds, UK
[4]Department of Health Services Research and Policy, London School of Hygiene & Tropical Medicine, London, UK
[5]Parent representative, Sussex, UK
[6]Paediatric Intensive Care Unit, Birmingham Women's and Children's Hospitals NHS Foundation Trust, Birmingham, UK
[7]Paediatric Intensive Care Unit, Great Ormond Street Hospital For Children NHS Trust, London, UK
[8]UCL Great Ormond Street Institute of Child Health, University College London, London, UK
[9]School of Health and Society, University of Salford, Salford, UK
[10]Children's Acute Transport Service, Great Ormond Street Hospital For Children NHS Trust, London, UK

**Acknowledgements** The authors thank Karen Thomas, Izabella Orzechowska, Michelle Saull and Roger Parslow for their contribution to the set-up and delivery of FIRST-ABC. The authors also thank the research and clinical staff at the participating sites: Addenbrookes' Hospital (Cambridge), Alder Hey Children's Hospital (Liverpool), Birmingham Women and Children's Hospital, Bristol Royal Hospital for Children, Chelsea and Westminster Hospital, Evelina London Children's Hospital, Great North Children's Hospital (Newcastle), Great Ormond Street Hospital (London), Hull Royal Infirmary, James Cook University Hospital (Middlesbrough), John Radcliffe Hospital (Oxford), King's College Hospital (London), Leicester Royal Infirmary and Glenfield Hospital, Noah's Ark Children's Hospital for Wales (Cardiff), Queens Medical Centre (Nottingham), Royal Alexandra Children's Hospital

(Brighton), Royal Brompton Hospital (London), Royal Hospital for Sick Children Edinburgh, Royal Manchester Children's Hospital, Sheffield Children's Hospital, Southampton Children's Hospital, St George's Hospital (London), St Mary's Hospital (London) and The Royal London Hospital. The authors acknowledge the UK Paediatric Intensive Care Society—Study Group for supporting the trial.

**Contributors** PR is chief investigator. AR-B is trial manager. PR and AR-B drafted the manuscript. PD, RF, RG, DAH, JL, KMR, PRM, MJP, KMR, ZS, and LNT are trial coapplicants and members of the Trial Management Group. LD supported management of the trial. All authors read and approved the final version.

**Funding** This trial is funded by the National Institute for Health Research (NIHR) Health Technology Assessment (HTA) Programme (project number: 17/94/28). Great Ormond Street Hospital for Children NHS Foundation Trust is the trial sponsor.

**Disclaimer** The views expressed are those of the author(s) and not necessarily those of the NIHR or the Department of Health and Social Care nor the Sponsor. The funder and sponsor had no role in the writing of this article or the decision to submit the protocol for publication.

**Patient and public involvement** Patients and/or the public were involved in the design, or conduct, or reporting, or dissemination plans of this research. Refer to the Methods section for further details.

**Patient consent for publication** Not required.

**Provenance and peer review** Not commissioned; externally peer reviewed.

**ORCID iDs**
Alvin Richards-Belle http://orcid.org/0000-0001-8577-9380
Padmanabhan Ramnarayan http://orcid.org/0000-0003-0784-8154

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
