## [Reviewer comments · BMJ Open]

ARTICLE DETAILS

TITLE (PROVISIONAL)	FIRST-line support for Assistance in Breathing in Children (FIRST-ABC): A master protocol of two randomised trials to evaluate the non-inferiority of high flow nasal cannula (HFNC) versus continuous positive airway pressure (CPAP) for non-invasive respiratory support in paediatric critical care
AUTHORS	Richards-Belle, Alvin; Davis, Peter; Drikite, Laura; Feltbower, Richard; Grieve, Richard; Harrison, David; Lester, Julie; Morris, Kevin; Mouncey, Paul; Peters, Mark; Rowan, Kathryn; Sadique, Zia; Tume, Lyvonne; Ramnarayan, Padmanabhan

VERSION 1 – REVIEW

REVIEWER	Vicent Modesto i Alapont PICU. Hospital Universitari i Politècnic La Fe. València. Spain
REVIEW RETURNED	24-Mar-2020

GENERAL COMMENTS	I think this protocol is rigorous, and both trials must be done. Their results will be extremely relevant to the clinical practice all over the world. Both are necessary trials.
---

REVIEWER	Fulya Kamit Istanbul Gaziosmanpasa Hospital, Yenyuzuyil University, Istanbul
REVIEW RETURNED	12-Apr-2020

GENERAL COMMENTS	To the authors, It is my pleasure to review this important RCT protocol evaluating the clinical and costeffectivity of HFNC treatment (comparing to CPAP), a noninferiority study. I have serious concerns about the protocols both for CPAP and HFNC, inclusion criteria and primary outcome. I marked them on the text. Yours Sincerely The reviewer provided a marked copy with additional comments. Please contact the publisher for full details.
---

REVIEWER	Andrea Wolfler Children's Hospital Vittore Buzzi
REVIEW RETURNED	25-Apr-2020

GENERAL COMMENTS	Dear Editor, I read with interest the paper entitled FIRST-line support for Assistance in Breathing in Children (FIRST-ABC). The paper is well written and the protocol is clear and pragmatic, addressing a useful topic in the daily PICU work. I have few minor revisions.
--

	Hypercapnia might coexist with a normal oxygen saturation and apparent patient comfort. In the treatment failure of both the two studies, why the authors did not include hypercapnia as a possible criteria to be considered ? Or is it included in the respiratory distress evaluation ? Does the authors allow any kind of sedation during NRS ? Patient discomfort as reported in the study algorithm might be treated with light sedation before be considered as treatment failure ? In the study protocol the authors do not specify how children should be fed during NRS. Respiratory distress can reduce the ability to feed and at the same time an NG tube could be less tolerated during HFNC or CPAP. On the other hand, a prolonged starvation or reduced caloric intake could weaken the child. Do the authors think to collect some data about this topic and analyse this variable ? I think that among so many PICUs (25) experience with NRS might be different. Do the authors consider to define and/or to rate the confidence each center has, before the trial, with this kind of support ? Especially CPAP with some interfaces requires skills from physician and nurses. Otherwise this could be a confounding factor. I did not find any inclusion criteria about the respiratory distress (i.e. specific clinical values) to start with NRS. Do the authors specify these information with the participating center or is the inclusion just after a clinical evaluation by the treating clinicians ?
--	---

REVIEWER	Reinout A. Bem Amsterdam University Medical Centers, location AMC, Amsterdam, The Netherlands
REVIEW RETURNED	04-May-2020

GENERAL COMMENTS	In this work, Richards-Belle and colleagues report their master protocol for the multi-center FIRST-ABC trial, which aims to determine the clinical and cost-effectiveness of high flow nasal cannula (HFNC) as compared to continuous positive airway pressure (CPAP) in the pediatric intensive care unit (PICU). This is a pragmatic trial and consists of two non-inferiority RCT's, which both evaluate HFNC compared to CPAP as a first-line non-invasive respiratory support modality: a step-up RCT in the acute phase of respiratory failure, and a step-down RCT following extubation from invasive ventilation. This trial follows a large pilot-RCT as published previously by Ramnarayan et al. in Critical Care 2018 (doi.org/10.1186/s13054-018-2080-3). The manuscript, including the study methodology, is written well, and the study is performed by a group of experienced investigators. The combination of two RCT's, pertaining to two different clinical scenarios, within the same trial infrastructure is a clear strength of the study. The research question under study is certainly relevant, as the popularity and use of HFNC in the PICU has been increasing in the last decade without proper scientific evidence for its effectiveness. Although several studies have shown that HFNC has some level of efficacy in the treatment of oxygenation and/or ventilation failure in children, in combination with overall well tolerability, the effectiveness of HFNC for clinically relevant outcomes in daily practice settings remains to be determined. The
---

	information derived from this trial will be relevant mostly for PICU clinicians. As stated on page 20 (Trial status), this study is ongoing and recruitment started in August 2019. Planned completion dates are Nov 2020 and Jan 2022 for the step-down and step-up RCT, respectively. As such, I understand the authors have limited possibilities to respond to some of the comments below. MAJOR: 1) My first comment is related to the primary outcome. This study is designed as a pragmatic trial to evaluate effectiveness of HFNC vs CPAP in a daily clinical practice (“real world”) setting. The pragmatic design can be a clear strength of the study, however, to me, the chosen primary outcome (time to liberation from any form of respiratory support for 48hr), is less pragmatic. I believe a main reason for PICU clinicians to start non-invasive respiratory support (either HFNC or CPAP) in daily practice is to avoid endotracheal intubation and invasive ventilation, which may be associated with more complications and patient discomfort. As a clinician, I would be much more interested in whether HFNC (or CPAP) avoids invasive ventilation as a primary outcome. Considering the scenario of a child entering the step-up RCT on either HFNC or CPAP, but then failing this modality to be intubated for invasive ventilation: how will the authors be able to properly correct for the potentially large influence of events and complications during the subsequent period of invasive ventilation, which may cause large bias to the primary outcome of time to liberation from any form of respiratory support? Also, in case of need for non-invasive respiratory support after extubation in children who failed HFNC/CPAP in the step-up RCT, any choice during this step-down period between HFNC or CPAP, being the interventions under study in the step-up, will influence the primary outcome (note that children cannot enter both the step-up and step-down RCT, page 7, exclusion criteria). Finally, the authors mention the primary outcome was informed by the pilot FIRST-ABC trial (Ramnarayan et al. in Critical Care 2018, doi.org/10.1186/s13054-018-2080-3). However, in the reporting of that pilot trial, time to liberation from any form of respiratory support was not provided (see Table 3 of the pilot trial publication). I believe a more thorough description of the decision process and reasons for choosing the specific primary outcome may benefit the manuscript. This appears relevant in particular for pragmatic trial reporting. 2) One of the secondary outcomes is “proportion of patients in whom sedation is used” (page 14). Please consider to include more specifically the type of medications and (cumulative) dosing, as with regard to a comparison between HFNC and CPAP this may be far more important and informative. Also related to this point: how will the authors assess for usage and (prolonged) effects of sedatives during invasive ventilation in the step-down RCT? Drug withdrawal and/or delirium may be an important cause of increased respiratory support necessitating non-invasive respiratory support or re-intubation.
--	---

3) Please consider adding important secondary outcomes related to respiratory physiology, e.g. change in respiratory rates, blood gas analysis, SF ratios etc.

4) According to the pragmatic trial design, the proposed algorithms (Figure 1 and 2) for decisions regarding treatment escalation or weaning etc, are widely interpretable. However, the authors mention they used "Pre-specified objective criteria..." (line 31, page 10) in these algorithms, but statements as "mild", "severe" respiratory distress or patient comfort as used in the flow diagrams are hardly objective. Why did the authors only include some guidance regarding FiO2 in these algorithms, and not for pH/pCO2? In my opinion, ventilation failure is an important reason for starting non-invasive respiratory support such as HFNC for PICU clinicians (maybe even more than oxygenation failure), and thus should be included in the algorithm if FiO2 is also included. In line with this, why did the authors not include SF ratios instead of only FiO2?

In addition, it will be important to state whether the different participating centers already have local protocols in place for starting non-invasive respiratory support such as HFNC and CPAP. As the proposed algorithms leave much room for interpretation (again, which I can certainly appreciate given the pragmatic trial design), it is possible that local site protocols influence decision making, and thus this will have consequences for the generalizability of the results.

MINOR:

1) As mentioned by the authors in the Introduction (2nd paragraph, page 5), the evidence for CPAP, although widely used historically, is limited. Thus, in a most strict sense, this trial investigates non-inferiority of HFNC compared to an unproven non-invasive respiratory support modality. Given the great effort of setting-up this large multicenter trial, did the authors consider adding a third group with a first-line standard (low flow/non-rebreathing mask) oxygen strategy? I fully understand that currently this is not the primary focus of this study, but I believe this issue deserves thorough discussion in the final, future reporting of the results of this ongoing trial.

2) In general a drawback of pragmatic trial design as compared to for example a pure explanatory trial, is larger variability due to heterogeneity that may be associated with this 'real world simulation', requiring larger sample size. Should this be considered/accounted for in the sample size calculation (page 16 and 17)?

3) On page 16, the authors refer to reference 30 for CONSORT extension for non-inferiority trials. Please consider to implement and refer to the CONSORT extension for pragmatic trials by Zwarenstein M. et al. in the BMJ (2008, volume 337).

4) Page 20 describes the Patient/Public involvement. One parent was included as a co-investigator as a co-author of this manuscript. Data from parental reporting of the pilot trial was used to inform the current trial. Can the authors elaborate a little bit more on the process of parent participation (e.g. was there a formal parent advisory board)? It may be valuable for a pragmatic

	trial to more explicitly state which issues were brought up/found to be most important to the public/parents, and subsequently incorporated in the study design. This way the value of PP advisory committees is highlighted. 5) On page 6 (line 33), the authors mention that feedback from clinicians was used to inform the study design. Can the authors elaborate on how this feedback was provided? E.g. was this a survey among the participants? 6) Page 7 reports the exclusion criteria. Please note that bullet 1 does not explicitly state the inability to maintain a patent airway due to reduced level of consciousness. In addition, there is no mentioning of patients with recent gastric or esophageal surgery or esophageal atresia and possible risk of non-invasive respiratory support. Also, what about children with (severe) hemodynamic instability? Finally, please more explicitly state/give examples for "other forms of non-invasive respiratory support" (last bullet): e.g. bi-level positive pressure forms. 7) Looking at the situation in my own country, many eligible patients will have received HFNC in a community, non-academic, hospital prior to PICU transfer, and this may often be more than 2 hours (exclusion criterium, page 7). Can the authors reflect on this issue? 8) In the algorithms of Figure 1 and 2 one of the boxes states "to other forms of non-invasive AND/OR invasive ventilation". I do not understand the use of "AND". 9) Related to weaning patients from HFNC: in my experience flows are frequently decreased to well below a level at which any form of 'real' non-invasive respiratory support can be expected. Reduction of flow towards low flow levels on HFNC has the advantage to keep the child on the same cannula system for patient comfort. Is this something that the authors need to address (e.g. HFNC flow threshold at which it is called 'respiratory support' instead of 'oxygen support' only)? This may be relevant as the primary outcome is time to liberation from any form of respiratory support, excluding supplemental oxygen support alone. 10) Please explain better to the readers the reason for randomization stratification by age (line 4, page 8).
--	---

REVIEWER	Dr Sanja Zivanovic University of Oxford, UK
REVIEW RETURNED	09-May-2020

GENERAL COMMENTS	Well designed trial, with pragmatic approach, especially with regards to the attempt of answering 2 important questions in one trial Points to clarify: 1. The choice of primary outcome would benefit from more clarification prior to publication. Most of the published trials investigating modes of NIV support have been calculated on the (re)intubation rate. The authors state that this trial is trying to answer an "important clinical dilemma", (page 5, paragraph 3, line 4) and then proceed to explain that the
---

	primary outcome will be liberation from NIV at 48 hours post extubation. As a clinician, I am sure many of my colleagues will agree, the intubation (re-intubation) would be much more significant event than another day (or more hours) on the NIV. Please can you elaborate further on the choice of primary outcome, in particular why it was decided not to choose intubation/re-intubation. 2. Another potential problem is the heterogeneity of PICU patients, age groups and co-morbidities. With regards to analysis the binary division of age might not be sufficient, as the range from 13 months to 16 years of thoracic growth and therefore the subsequent FRC is very wide, and will differ greatly within the group of patient >12 months. It will greatly vary with onset of puberty as well. Age could be taken into analysis as an independent variable, when sub-analyzing the outcome in the older group (>12 mo). Onset of puberty should have been taken into consideration. Apart from neuromuscular co-morbidity, chronic lung disease should have been taken into account separately (not with the "other" group) 3. The practical issue of a center's familiarity with a particular mode of NIV is significant. If they have been using one mode for several years, training for the other mode should have been provided for the nursing staff. Please explain the decision not to. 4. In the step down arm, there is a very long time to start the NIV post extubation. The success of NIV might vary according to how soon after extubation it was started, to prevent atelectasis. Will the clinician's decision be clarified in the CRF? If NIV was started prophylactically, and child randomized prior to extubation, or were they extubated to air/NC oxygen and monitored for deterioration, prior the NIV was started? The time after extubation the NIV was started might be significant and should be recorded, as well as the clinician's reasoning for starting (prophylactic vs rescue) 5. With regards to the quality of life questionnaire after discharge, it was not explained in the introduction how does this relate to the choice of NIV? Could this please be added to intro. 6. With regards to consent, it is stated that in the case of consent not obtained, the data will still be used. This is unusual. It is clear that the delayed consent is necessary. However, if not obtained after 2 attempts (letter or phone call), the data should not be used. Otherwise the whole purpose of consenting for research studies is completely bypassed. Please explain.. 7. Last point, is the starting level of flow for the HF group. Starting the HF for infants larger than 3.5 kg will give HF of 7 L/min, which gives approximately equivalent PEEP to CPAP of 7 cmH₂O (Wilkinson et al, "Pharyngeal pressure with high-flow nasal cannulae in premature infants", Journal of perinatology, 2008). If, on the other hand, the child is smaller, i.e just above 2 – 2.5kg, this might make clinicians start the HF at 4-5 l/min, per this study protocol. This would make the HF group disadvantaged in comparison to the amount of PEEP it provides. Therefore, infants with the admission weight below 3.5 kg should be excluded from analysis, or lowest starting HF rate should be 7 l/min.
--	---

VERSION 1 – AUTHOR RESPONSE

Reviewer 1

I think this protocol is rigorous, and both trials must be done. Their results will be extremely relevant to the clinical practice all over the world. Both are necessary trials.

Thank you for this comment.

Reviewer 2

There is one RCT (TRAMONTANE study) evaluating efficacy and safety of HFNC compared to CPAP (as initial respiratory support) in patients with moderate to severe bronchiolitis in PICU (p. 6)

The TRAMONTANE trial was conducted only in Infants with bronchiolitis. We acknowledge and referenced this in the Introduction where we state 'There are few RCTs comparing HFNC with CPAP in the PICU setting. Previous RCTs do not include children with a range of ages and diagnoses needing either step-up or step-down (post-extubation) care, making it impossible to generalise their findings to contemporary practice.'

It is a large range of ages and diagnoses for respiratory diseases. It is better to separate the age groups and the diagnoses. It is well known that HFNC works well in patients with bronchiolitis (under 12 months of age) but sepsis in adolescent patients with comorbidities like hematologic malignancies? It is not ethical, you should define the diagnoses (at least pulmonary, nonpulmonary...) and severity of respiratory distress (p. 6)

Both HFNC and CPAP are already used in clinical practice across the large range of ages and diagnoses in the PICU. FIRST-ABC is a pragmatic study which aims to provide evidence directly relevant for contemporary PICU practice. We agree with the reviewer that there will be a wide range of ages and diagnoses included in the study, therefore we have planned subgroup analyses to study the effect size of the intervention in specific populations. These subgroups include age (<12 months versus ≥12 months), admission diagnosis, severity of respiratory distress and SF ratio among other subgroups (please see page 15 of the manuscript for full details). As stated in the manuscript, FIRST-ABC received favourable ethical opinion from the research ethics committee and acceptability of the trial was informed by a successfully conducted prior pilot study.¹

Acute illness should be well defined or the type of respiratory illness or the severity of respiratory distress (p. 6)

The inclusion criteria are deliberately pragmatic; patients are eligible if the treating clinician determines that they need non-invasive respiratory support. In this way, the study population aims to reflect the patients that would receive HFNC or CPAP in usual clinical practice. The trial is designed as a pragmatic clinical trial. By specifying particular types of respiratory illness or severity of respiratory distress, we may potentially improve the internal validity of the study, but we will lose the ability to generalise the results to usual clinical practice, which may be very different if such inclusion criteria were used. We have added this sentence in the Design section (Page 6): "The pragmatic study design ensures that research findings can be more easily generalised to real-world practice". Data on comorbidities and severity of respiratory distress are collected and will be reported.

You should define how you will use HFNC or CPAP for postextubation, is it routine or for patients who need respiratory support after postextubation (level of respiratory distress) (p. 6)

This was defined in inclusion criteria 3B listed in the manuscript. See also the above response on the pragmatic nature of the inclusion criteria.

the primary outcome is not clear, if you are evaluating the effectiveness first primary outcome would

be success or failure, then timing (both for efficacy and costeffectiveness). (p. 6)
Treatment 'success' or 'failure' has been used in previous trials, largely as composite outcome measures which have been difficult to translate into clinical practice. Moreover, use of these outcomes does not account for what happens after the initial 'failure'. We have chosen a primary outcome that was preferred by clinicians as well as parents/families. See additional text on page 12 related to choice of the primary outcome.

please give reference for this (p. 9)
Done.

please define the type of PICUs (cardiac-medical-surgical, tertiary or secondary) (p. 9)
The participating PICUs are a mix of general medical-surgical, cardiac and mixed units. This detail has been added to the manuscript. Nearly all PICUs in the UK are participating in the trial.

Admission criterias are very different between units regarding the diagnoses, you should define it clearly (moderate to severe for bronchiolitis, severe for pneumonia; or severity of respiratory distress (vital signs or FiO2 need...) (p. 10)

Please see responses above. The inclusion criteria are deliberately pragmatic; patients are eligible if the treating clinician determines that they need non-invasive respiratory support. In this way, the study population aims to reflect the patients that would receive HFNC or CPAP in usual clinical practice. When the study is reported we will present data on the types of illness that were recruited to the study, as well as severity of illness at randomisation.

You should clearly define the requirement of NIRS (decision) according to ages and diagnoses because of practices are very different among units regarding the treatment decisions, not uniform. (p. 10)

The inclusion criteria are deliberately pragmatic; patients are eligible if the treating clinician determines that they need non-invasive respiratory support. In this way, the study population aims to reflect the patients that would receive HFNC or CPAP usual clinical practice.

Please indicate the definitions FOR indication (oxygen need, respiratory distress) for NIRS, deterioration, severe respiratory distress??????? (p. 12)

As mentioned above, the need for non-invasive respiratory support is determined pragmatically by treating clinicians. We have added the definitions for levels of respiratory distress in the footnotes for Figures 1 and 2.

You should clearly define the clinical response by vital signs (RR reduction, FiO2 need reduction.....) (p. 12)

The wide range of ages and diagnoses included in the trial does not allow easy criteria by which clinical response can be judged in simple numbers. We have provided definitions of treatment failure in the algorithms, which will allow clinicians to judge the success or failure of the treatment.

You should clearly define no response according to ages and diagnoses. Failure is different while using HFNC for mild bronchiolitis, sepsis or PARDS at risk or PARDS. (p. 12)

The wide range of ages and diagnoses included in the trial does not allow easy criteria by which clinical response can be judged in simple numbers. We agree with the reviewer that there will be a wide range of ages and diagnoses included in the study, therefore we have planned subgroup analyses to study the effect size of the intervention in specific populations. These subgroups include age (<12 versus \geq 12 months), admission diagnosis, severity of respiratory distress and SF ratio among other subgroups (see manuscript page 15 for full details). We have provided definitions of treatment failure in the algorithms, which will allow clinicians to judge the success or failure of the treatment.

You should define the timing for assessing the critically ill child for response (p. 12)

Timing of assessment and determination of success/failure is left to clinical discretion by treating team. We state in the manuscript that patients should be assessed for response at least twice per day (see pages 8 and 10).

How will you titrate the dose (increasing/decreasing flows, FiO₂) by time? (p. 12)

The algorithm specifies how the flow rate should be decreased if the criteria for success are met.

Criteria are not clear exactly in the algorithms, especially while treating wide range of critically ill children in PICU. (p. 13)

The algorithms were developed in consultation with clinicians from paediatric critical care across the UK and feedback and implementation has indicated that they are clear. Clinicians indicated that algorithms needed to allow for clinical discretion rather than being too prescriptive, and it was also important the algorithms reflected current practice.

Do you have a sedation protocol for both HFNC and CPAP. It is needed if you are assessing the patient discomfort. (p. 14)

No sedation protocol is provided for the trial – individual clinicians will continue usual practice. Both use of sedation and patient comfort are recorded and will be reported as a secondary outcome.

Will you switch to HFNC if CPAP fails? The algorithms for HFNC and CPAP is not clear and also risky: You use the almost the same protocol both for HFNC and CPAP while evaluating the noninferiority or efficacy for HFNC comparing to CPAP. It is not right. (p. 14)

The algorithms were developed in consultation with clinicians from paediatric critical care across the UK and feedback and implementation has indicated that they are clear. Clinicians have the option to change from one treatment to the other, or to other forms of respiratory support, at their discretion. See additional text on Page 8 regarding the development of the algorithms.

It is not clear at all. What is the definition of free of all forms of respiratory support? Will the patient stay in PICU or transport to ward setting? (p. 17)

We have added text to clarify that this outcome is measured even after discharge from critical care.

what is your hypothesis? If you have, it is better to indicate it. (p. 19)

The hypothesis is: In critically ill children assessed by the treating clinician to require non-invasive respiratory support (NRS), first-line use of high flow nasal cannula (HFNC) is non-inferior to continuous positive airway pressure (CPAP) in time to liberation from respiratory support. This has been added to the Methods section.

Reviewer 3

Dear Editor, I read with interest the paper entitled FIRST-line support for Assistance in Breathing in Children (FIRST-ABC). The paper is well written and the protocol is clear and pragmatic, addressing a useful topic in the daily PICU work. I have few minor revisions.

Thank you for this comment. We address your queries below.

Hypercapnia might coexist with a normal oxygen saturation and apparent patient comfort. In the treatment failure of both the two studies, why the authors did not include hypercapnia as a possible criteria to be considered? Or is it included in the respiratory distress evaluation?

Assessment of hypercapnia requires a blood gas analysis, which we found was not universally performed to assess treatment failure in the pilot RCT.¹ Clinicians assessed success/failure by clinical grounds alone, including FiO₂ and degree of respiratory distress. In order to provide a simple and non-invasive measure by which failure/success is assessed in every case, we chose not to

include hypercapnia in the definition of treatment failure. In practice, analysis of the pilot RCT data¹ showed that hypercapnia (where available) was always correlated with respiratory distress.

Does the authors allow any kind of sedation during NRS? Patient discomfort as reported in the study algorithm might be treated with light sedation before be considered as treatment failure ?

This is true – sedation practices are as per usual unit policies. The trial did not specify any sedation protocols. See page 9 for clarification regarding sedation practices.

In the study protocol the authors do not specify how children should be fed during NRS. Respiratory distress can reduce the ability to feed and at the same time an NG tube could be less tolerated during HFNC or CPAP. On the other hand, a prolonged starvation or reduced caloric intake could weaken the child. Do the authors think to collect some data about this topic and analyse this variable ?

We did not specify a feeding protocol in the trial, instead allowing usual practice to continue. However unfortunately we did not collect data on feeding practices. See page 9 for clarification regarding feeding practices.

I think that among so many PICUs (25) experience with NRS might be different. Do the authors consider to define and/or to rate the confidence each center has, before the trial, with this kind of support ? Especially CPAP with some interfaces requires skills from physician and nurses. Otherwise this could be a confounding factor.

We agree with this comment. During the preparatory phase of the trial, we ascertained from sites their usual practices for NIV and HFNC usage. Bar two PICUs, all used both treatments in routine practice. The two PICUs that used NIV infrequently undertook a staff education program on the unit prior to recruitment. This is also why for the protocol does not impose a single system or interface but allows sites to use the method of delivery/interface for both CPAP and HFNC that they were already familiar with using.

I did not find any inclusion criteria about the respiratory distress (i.e. specific clinical values) to start with NRS. Do the authors specify these information with the participating center or is the inclusion just after a clinical evaluation by the treating clinicians ?

It is the latter, i.e. after clinical evaluation by the treating clinician.

Reviewer 4

In this work, Richards-Belle and colleagues report their master protocol for the multi-center FIRST-ABC trial, which aims to determine the clinical and cost-effectiveness of high flow nasal cannula (HFNC) as compared to continuous positive airway pressure (CPAP) in the pediatric intensive care unit (PICU). This is a pragmatic trial and consists of two non-inferiority RCT's, which both evaluate HFNC compared to CPAP as a first-line non-invasive respiratory support modality: a step-up RCT in the acute phase of respiratory failure, and a step-down RCT following extubation from invasive ventilation. This trial follows a large pilot-RCT as published previously by Ramnarayan et al. in *Critical Care* 2018 (doi.org/10.1186/s13054-018-2080-3). The manuscript, including the study methodology, is written well, and the study is performed by a group of experienced investigators. The combination of two RCT's, pertaining to two different clinical scenarios, within the same trial infrastructure is a clear strength of the study. The research question under study is certainly relevant, as the popularity and use of HFNC in the PICU has been increasing in the last decade without proper scientific evidence for its effectiveness. Although several studies have shown that HFNC has some level of efficacy in the treatment of oxygenation and/or ventilation failure in children, in combination with overall well tolerability, the effectiveness of HFNC for clinically relevant outcomes in daily practice settings remains to be determined. The information derived from this trial will be relevant mostly for PICU clinicians. As stated on page 20 (Trial status), this study is ongoing and recruitment started in August 2019. Planned completion dates are Nov 2020 and Jan 2022 for the step-down and step-up RCT, respectively. As such, I understand the authors have limited possibilities to respond to some of the

comments below.

Thank you for these comments.

1) My first comment is related to the primary outcome. This study is designed as a pragmatic trial to evaluate effectiveness of HFNC vs CPAP in a daily clinical practice (“real world”) setting. The pragmatic design can be a clear strength of the study, however, to me, the chosen primary outcome (time to liberation from any form of respiratory support for 48hr), is less pragmatic. I believe a main reason for PICU clinicians to start non-invasive respiratory support (either HFNC or CPAP) in daily practice is to avoid endotracheal intubation and invasive ventilation, which may be associated with more complications and patient discomfort. As a clinician, I would be much more interested in whether HFNC (or CPAP) avoids invasive ventilation as a primary outcome. Considering the scenario of a child entering the step-up RCT on either HFNC or CPAP, but then failing this modality to be intubated for invasive ventilation: how will the authors be able to properly correct for the potentially large influence of events and complications during the subsequent period of invasive ventilation, which may cause large bias to the primary outcome of time to liberation from any form of respiratory support? Also, in case of need for non-invasive respiratory support after extubation in children who failed HFNC/CPAP in the step-up RCT, any choice during this step-down period between HFNC or CPAP, being the interventions under study in the step-up, will influence the primary outcome (note that children cannot enter both the step-up and step-down RCT, page 7, exclusion criteria). Finally, the authors mention the primary outcome was informed by the pilot FIRST-ABC trial (Ramnarayan et al. in *Critical Care* 2018, doi.org/10.1186/s13054-018-2080-3). However, in the reporting of that pilot trial, time to liberation from any form of respiratory support was not provided (see Table 3 of the pilot trial publication). I believe a more thorough description of the decision process and reasons for choosing the specific primary outcome may benefit the manuscript. This appears relevant in particular for pragmatic trial reporting.

The choice of primary outcome involved considerable deliberation and discussion within the research team as well as outside the research team. Our primary outcome was informed by discussions with clinicians as well as parents/families. The main reasons for choosing the ‘time to liberation from respiratory support’ outcome were:

1) Parents/families reported that even though intubation was clearly an undesirable outcome, the fact that the child needed a ‘breathing machine’ of any description would be more important for them, in terms of assessing the success or failure of the intervention. Normalisation of ‘breathing’ was an important outcome prioritised over intubation.

2) Since the rate of intubation on average was around 20% (see pilot RCT), nearly 80% of patients do not fulfil the intubation outcome.¹ In these patients, several non-invasive support modes may be used, which prolong the time the patient is on ‘breathing support’ with resource implications for critical care. Clinicians felt that it was important that the effect of the intervention was assessed on patients who did not need intubation as well as on those who did.

3) In the pilot RCT (table 3), we present data on the ventilator free days at day 28, which included non-invasive support.¹ We mention in the Discussion section: “Furthermore, and irrespective of whether patients were switched or escalated to other treatments, the choice of first-line NRS mode influenced the rate of intubation and overall length of respiratory support, indicating that they might be candidate outcome measures for a future RCT”.

4) Unpublished data from the pilot RCT showed that the length of respiratory support is longer in patients who need intubation compared to those who do not. Therefore, the adverse impact of intubation is likely reflected in longer duration of respiratory support anyway.

5) The vast majority of centres participating in FIRST-ABC trial have just completed participation in the SANDWICH trial,² which provided protocolised management of weaning/extubation in invasively ventilated patients, which would have allowed for standardisation of practice across sites.

6) We agree that a large number of events may occur after intubation which affect the primary outcome, but we do not expect these to affect CPAP and HFNC randomised patients differently.

Randomisation is also stratified by site, therefore minimising any differences in practice between

sites. We have therefore added further detail to the Outcome measures section to provide some of the context behind the primary outcome choice.

One of the secondary outcomes is “proportion of patients in whom sedation is used” (page 14). Please consider to include more specifically the type of medications and (cumulative) dosing, as with regard to a comparison between HFNC and CPAP this may be far more important and informative. Also related to this point: how will the authors assess for usage and (prolonged) effects of sedatives during invasive ventilation in the step-down RCT? Drug withdrawal and/or delirium may be an important cause of increased respiratory support necessitating non-invasive respiratory support or re-intubation.

Secondary outcomes are pre-specified and will be reported as per the protocol. In order to minimise data collection burden, we are not collecting the dose and types of sedative medication used, however, we have defined the agents that fit the definition of “sedation” as part of the case report form. This will be reported.

Please consider adding important secondary outcomes related to respiratory physiology, e.g. change in respiratory rates, blood gas analysis, SF ratios etc.

Data on respiratory physiology are collected as part of the hourly data in the first 6 hours and 6 hourly data subsequently. These will be reported. Secondary outcomes are pre-specified and will be reported as per the protocol.

According to the pragmatic trial design, the proposed algorithms (Figure 1 and 2) for decisions regarding treatment escalation or weaning etc, are widely interpretable. However, the authors mention they used “Pre-specified objective criteria....” (line 31, page 10) in these algorithms, but statements as “mild”, “severe” respiratory distress or patient comfort as used in the flow diagrams are hardly objective. Why did the authors only include some guidance regarding FiO₂ in these algorithms, and not for pH/pCO₂? In my opinion, ventilation failure is an important reason for starting non-invasive respiratory support such as HFNC for PICU clinicians (maybe even more than oxygenation failure), and thus should be included in the algorithm if FiO₂ is also included. In line with this, why did the authors not include SF ratios instead of only FiO₂? In addition, it will be important to state whether the different participating centers already have local protocols in place for starting non-invasive respiratory support such as HFNC and CPAP. As the proposed algorithms leave much room for interpretation (again, which I can certainly appreciate given the pragmatic trial design), it is possible that local site protocols influence decision making, and thus this will have consequences for the generalizability of the results.

We have added the definitions for levels of respiratory distress in the footnotes for Figures 1 and 2. Assessment of hypercapnia requires a blood gas analysis, which we found was not universally performed to assess treatment failure in the pilot RCT.¹ Clinicians assessed success/failure by clinical grounds alone, including FiO₂ and degree of respiratory distress. In order to provide a simple and non-invasive measure by which failure/success is assessed in every case, we chose not to include hypercapnia in the definition of treatment failure. In practice, analysis of the pilot RCT data showed that hypercapnia (where available) was always correlated with respiratory distress.¹

The guidance for saturation targets are included in the algorithm and specify a target of >92% (unless a patient specific target alteration is needed). We will be reporting SF ratios since we will have data on FiO₂ and SpO₂. We agree that units may have different thresholds for starting non-invasive support, although this variability may be more clinician-specific rather than centre-specific. Randomisation will be stratified by site, however.

As mentioned by the authors in the Introduction (2nd paragraph, page 5), the evidence for CPAP, although widely used historically, is limited. Thus, in a most strict sense, this trial investigates non-inferiority of HFNC compared to an unproven non-invasive respiratory support modality. Given the

great effort of setting-up this large multicenter trial, did the authors consider adding a third group with a first-line standard (low flow/non-rebreathing mask) oxygen strategy? I fully understand that currently this is not the primary focus of this study, but I believe this issue deserves thorough discussion in the final, future reporting of the results of this ongoing trial.

We agree with the comment that CPAP is also largely a non-evidence-based therapy and the risk of HFNC becoming the “new CPAP” is quite high if this trial were not conducted. However, adding a low flow oxygen strategy may not have been easily justifiable in a high-risk population considering that there is accumulating systematic review evidence that standard oxygen therapy may be associated with worse outcomes than HFNC, mainly in adults so far. We will be discussing this in the final report of the trial.

In general a drawback of pragmatic trial design as compared to for example a pure explanatory trial, is larger variability due to heterogeneity that may be associated with this ‘real world simulation’, requiring larger sample size. Should this be considered/accounted for in the sample size calculation (page 16 and 17)?

Please see the sample size section in the manuscript which confirms that allowances in the sample size are made, including “for exclusion due to non-adherence in the per-protocol population”.

On page 16, the authors refer to reference 30 for CONSORT extension for non-inferiority trials.

Please consider to implement and refer to the CONSORT extension for pragmatic trials by Zwarenstein M. et al. in the BMJ (2008, volume 337).

We have additionally referred to this CONSORT extension.³

Page 20 describes the Patient/Public involvement. One parent was included as a co-investigator as a co-author of this manuscript. Data from parental reporting of the pilot trial was used to inform the current trial. Can the authors elaborate a little bit more on the process of parent participation (e.g. was there a formal parent advisory board)? It may be valuable for a pragmatic trial to more explicitly state which issues were brought up/found to be most important to the public/parents, and subsequently incorporated in the study design. This way the value of PP advisory committees is highlighted.

Following the pilot RCT,¹ the Patient/Public Involvement Group for Research at Great Ormond Street Hospital was consulted on the choice of the primary outcome for the main RCTs. See comment above regarding primary outcome. Two parents were included as co-applicants on the initial grant application and continued to be on the research team when the trial started. Unfortunately, one parent was unable to continue their time commitment. The parent rep on the research team attends regular Trial Management Group meetings, and has been consulted on all participant information sheets and consent forms.

On page 6 (line 33), the authors mention that feedback from clinicians was used to inform the study design. Can the authors elaborate on how this feedback was provided? E.g. was this a survey among the participants? We have clarified which feedback we were referring to.

The trial algorithms were developed iteratively in consultation with paediatric critical care clinicians across the UK (both via email and in person at a Collaborators’ Meeting held prior to the start of the trial). We have added this detail to the manuscript.

Page 7 reports the exclusion criteria. Please note that bullet 1 does not explicitly state the inability to maintain a patent airway due to reduced level of consciousness. In addition, there is no mentioning of patients with recent gastric or esophageal surgery or esophageal atresia and possible risk of non-invasive respiratory support. Also, what about children with (severe) hemodynamic instability? Finally, please more explicitly state/give examples for “other forms of non-invasive respiratory support” (last bullet): e.g. bi-level positive pressure forms.

The first exclusion criterion states: Assessed by the treating clinician to require immediate intubation and invasive ventilation due to severe hypoxia, acidosis and/or respiratory distress, upper airway

obstruction, inability to manage airway secretions or recurrent apnoeas. Inability to manage airway secretions is used to refer to any situation where a patient is unable to manage their airway, whether from coma or from bulbar weakness. Gastric or oesophageal surgery patients are not excluded automatically, since in practice, many do receive non-invasive support. However, we recognise that clinicians at local level may not have equipoise in some groups of patients, therefore in rare situations, it is possible for the clinician to exercise their discretion and exclude patients from the trial. Such exclusions are monitored and discussed regularly within the research team. Other forms of non-invasive respiratory support includes: bilevel positive pressure, and negative pressure ventilation. We have added this clarification to the exclusion criteria.

Looking at the situation in my own country, many eligible patients will have received HFNC in a community, non-academic, hospital prior to PICU transfer, and this may often be more than 2 hours (exclusion criterium, page 7). Can the authors reflect on this issue?

We agree that this is the situation in the UK too. However, it would not be ethically possible to randomise patients who are either improving or not after the first couple of hours on their non-randomised treatment, therefore, these patients would be excluded. We would only be able to randomise patients who have just started their treatment.

In the algorithms of Figure 1 and 2 one of the boxes states “to other forms of non-invasive AND/OR invasive ventilation”. I do not understand the use of “AND”.

This wording acknowledges that some patients may be escalated to other forms of non-invasive ventilation and then to invasive ventilation (i.e. there can be more than one escalation).

Related to weaning patients from HFNC: in my experience flows are frequently decreased to well below a level at which any form of ‘real’ non-invasive respiratory support can be expected. Reduction of flow towards low flow levels on HFNC has the advantage to keep the child on the same cannula system for patient comfort. Is this something that the authors need to address (e.g. HFNC flow threshold at which it is called ‘respiratory support’ instead of ‘oxygen support’ only)? This may be relevant as the primary outcome is time to liberation from any form of respiratory support, excluding supplemental oxygen support alone.

Our weaning step is to reduce the flow to 1L/kg/min (or weight-banded equivalent flow rate), below which the flow rate would not be considered respiratory support. Therefore, the algorithm specifies that patients on 1 L/kg/min and stable can be taken off the HFNC and converted to supplemental oxygen (which is not considered respiratory support).

Please explain better to the readers the reason for randomization stratification by age (line 4, page 8). The reason (to minimise imbalance arising from unit practices and interface selection) for stratifying by site and age group is added to the Methods section (see page 8).

The choice of primary outcome would benefit from more clarification prior to publication.

Most of the published trials investigating modes of NIV support have been calculated on the (re)intubation rate. The authors state that this trial is trying to answer an “important clinical dilemma” , (page 5, paragraph 3, line 4) and then proceed to explain that the primary outcome will be liberation from NIV at 48 hours post extubation. As a clinician, an I am sure many of my colleagues will agree, the intubation (re-intubation) would be much more significant event than another day (or more hours) on the NIV.

Please can you elaborate further on the choice of primary outcome, in particular why it was decided not to choose intubation/re-intubation.

Our primary outcome was informed by discussions with clinicians as well as parents/families. The main reasons for choosing the ‘time to liberation from respiratory support’ outcome were:

- 1) Parents/families reported that even though intubation was clearly an undesirable outcome, the fact that the child needed a ‘breathing machine’ of any description would be more important for them, in

terms of assessing the success or failure of the intervention. Normalisation of 'breathing' was an important outcome prioritised over intubation.

2) Since the rate of intubation on average was around 20% (see pilot RCT), nearly 80% of patients do not fulfil the intubation outcome.¹ In these patients, several non-invasive support modes may be used, which prolong the time the patient is on 'breathing support' with resource implications for critical care. Clinicians felt that it was important that the effect of the intervention was assessed on patients who did not need intubation as well as on those who did.

3) In the pilot RCT (table 3), we present data on the ventilator free days at day 28, which included non-invasive support.¹ We mention in the Discussion section: "Furthermore, and irrespective of whether patients were switched or escalated to other treatments, the choice of first-line NRS mode influenced the rate of intubation and overall length of respiratory support, indicating that they might be candidate outcome measures for a future RCT".

4) Unpublished data from the pilot RCT showed that the length of respiratory support is longer in patients who need intubation compared to those who do not. Therefore, the adverse impact of intubation is likely reflected in longer duration of respiratory support anyway.

5) The vast majority of centres participating in FIRST-ABC trial have just completed participation in the SANDWICH trial,² which provided protocolised management of weaning/extubation in invasively ventilated patients, which would have allowed for standardisation of practice across sites.

6) We agree that a large number of events may occur after intubation which affect the primary outcome, but we do not expect these to affect CPAP and HFNC randomised patients differently. We have added some of this context to the 'Outcome measures' section.

Another potential problem is the heterogeneity of PICU patients, age groups and co-morbidities. With regards to analysis the binary division of age might not be sufficient, as the range from 13 months to 16 years of thoracic growth and therefore the subsequent FRC is very wide, and will differ greatly within the group of patient >12 months. It will greatly vary with onset of puberty as well. Age could be taken into analysis as an independent variable, when sub-analyzing the outcome in the older group (>12 mo). Onset of puberty should have been taken into consideration. Apart from neuromuscular co-morbidity, chronic lung disease should have been taken into account separately (not with the "other" group)

We anticipate that a high proportion (>50%) of patients will be in the <12-month age group. Accordingly, the expected number of patients who have reached puberty will be small, limiting exploration of this factor. We have added a sentence to confirm that we will explore using age as a continuous variable if there are sufficient patient numbers in the older age groups. The subgroup analyses will be reported as per the pre-specified definitions. Note that chronic lung diseases is included within the 'other respiratory' group for clinical indication subgroup in step-up RCT.

The practical issue of a center's familiarity with a particular mode of NIV is significant. If they have been using one mode for several years, training for the other mode should have been provided for the nursing staff. Please explain the decision not to.

During the preparatory phase of the trial, we ascertained from sites their usual practices for NIV and HFNC usage. Bar two PICUs, all used both treatments in routine practice. The two PICUs that used NIV infrequently undertook a local staff education program on the unit prior to recruitment.

In the step down arm, there is a very long time to start the NIV post extubation. The success of NIV might vary according to how soon after extubation it was started, to prevent atelectasis. Will the clinicians decision be clarified in the CRF? If NIV was started prophylactically, and child randomized prior to extubation, or were they extubated to air/NC oxygen and monitored for deterioration, prior the NIV was started? The time after extubation the NIV was started might be significant and should be recoded, as well as the clinicians reasoning for starting (prophylactic vs rescue)

We are collecting data on the time of extubation, randomisation and starting treatment. It will be possible to classify the step-down support into prophylactic and rescue based on the data. This will be

reported in the results.

With regards to the quality of life questionnaire after discharge, it was not explained in the introduction how does this relate to the choice of NIV? Could this please be added to intro.

The quality of life questionnaire is part of the health economic evaluation. The use of this data is described in the 'Integrated health economic evaluation' section.

With regards to consent, it is stated that in the case of consent not obtained, the data will still be used. This is unusual. It is clear that the delayed consent is necessary. However, if not obtained after 2 attempts (letter or phone call), the data should not be used. Otherwise the whole purpose of consenting for research studies is completely bypassed. Please explain.

The consent model used in FIRST-ABC has been directly informed by the CONSeNt methods in paediatric Emergency and urgent Care Trials (CONNECT) study guidance⁴ and conduct of several paediatric critical care trials conducted by our team, all of which have had extensive Patient and Public Involvement and engagement.

The "opt-out element" of the postal consent procedures was included based on recent evidence resulting from two qualitative interview studies conducted in the paediatric critical care setting (please see O'Hara et al., 2018⁵ and Peters et al., 2019⁶ references now added to the manuscript). Across the two studies, a total of 46 (13 bereaved) parents of critically ill children were interviewed. All parents found research without prior consent acceptable and indicated that they would have consented to use of their child's information in the respective trials under investigation (one was a trial of temperature thresholds and the other, fluid bolus therapy), had their child been enrolled using deferred consent.

We have evidence from the CATCH trial¹⁰, FiSh⁷ and CONNECT⁴ to believe that an alternative approach, such as a telephone call or 'opt-in' approach via a letter home is not supported by parents and will lead to missing data, which poses a significant threat to the validity of research and safety of participants. A published evaluation of the CATCH consent procedures¹¹ concluded that "RCTs should balance the potential burden that seeking consent in difficult situations may cause against risk of bias by disproportionately excluding children who die or are transferred."

Specifically, the majority of parents supported, and found acceptable, contact via post at 4 weeks and then at 8 weeks after death of their child, with a personalised 'opt-out' declaration - as long as the 'opt-out' declaration was emphasised in bold and the letter personalised, ideally by a known member of the clinical team. As this approach is deemed acceptable by parents, these consent procedures (including the personalised 'opt-out' element emphasised in bold), have now been used (and refined) in a number of recently completed paediatric critical care trials conducted by members of our team (FiSh⁷, Oxy-PICU⁸, Fever⁹, FIRST-ABC Pilot study¹), and supported by Patient and Public Involvement (parent) members of the respective Trial Management Groups.

We also highlight that the 'opt-out' element of the consent procedures is only considered as a last resort (i.e. there is a low mortality rate in paediatric critical care and we expect that for the vast majority of survivors, parents/legal guardians will be approached for consent in hospital) and this is emphasized in study procedures and site training.

We have added additional supporting references to the manuscript.

Last point, is the starting level of flow for the HF group. Starting the HF for infants larger than 3.5 kg will give HF of 7 L/min, which gives approximately equivalent PEEP to CPAP of 7 cmH₂O (Wilkinson et al, "Pharyngeal pressure with high-flow nasal cannulae in premature infants", Journal of perinatology, 2008). If, on the other hand, the child is smaller, i.e just above 2 – 2.5kg, this might

make clinicians start the HF at 4-5 l/min , per this study protocol. This would make the HF group disadvantaged in comparison to the amount of PEEP it provides. Therefore, infants with the admission weight below 3.5 kg should be excluded from analysis, or lowest starting HF rate should be 7 l/min.

There is no well validated approach to assessing the degree of PEEP provided by various flow rates in infants and children – variability in PEEP may result from mouth opening as well as flow rate and airway anatomy. Studies from Milesi et al. show that the average PEEP in bronchiolitis infants (not premature infants, who are not our study population) is around 4-5 cm H₂O for a flow rate of around 2 L/kg/min.¹² In addition, the beneficial effects of HFNC may not come solely from the PEEP, rather dead space washout may play a significant role too.

Please do not hesitate to get in touch if you have any further queries or need further information.

Reviewer 5

The choice of primary outcome would benefit from more clarification prior to publication. Most of the published trials investigating modes of NIV support have been calculated on the (re)intubation rate. The authors state that this trial is trying to answer an “important clinical dilemma” , (page 5, paragraph 3, line 4) and then proceed to explain that the primary outcome will be liberation from NIV at 48 hours post extubation. As a clinician, I am sure many of my colleagues will agree, the intubation (re-intubation) would be much more significant event than another day (or more hours) on the NIV. Please can you elaborate further on the choice of primary outcome, in particular why it was decided not to choose intubation/re-intubation.

Our primary outcome was informed by discussions with clinicians as well as parents/families. The main reasons for choosing the ‘time to liberation from respiratory support’ outcome were:

1) Parents/families reported that even though intubation was clearly an undesirable outcome, the fact that the child needed a ‘breathing machine’ of any description would be more important for them, in terms of assessing the success or failure of the intervention. Normalisation of ‘breathing’ was an important outcome prioritised over intubation.

2) Since the rate of intubation on average was around 20% (see pilot RCT), nearly 80% of patients do not fulfil the intubation outcome.¹ In these patients, several noninvasive support modes may be used, which prolong the time the patient is on ‘breathing support’ with resource implications for critical care. Clinicians felt that it was important that the effect of the intervention was assessed on patients who did not need intubation as well as on those who did.

3) In the pilot RCT (table 3), we present data on the ventilator free days at day 28, which included non-invasive support.¹ We mention in the Discussion section: “Furthermore, and irrespective of whether patients were switched or escalated to other treatments, the choice of first-line NRS mode influenced the rate of intubation and overall length of respiratory support, indicating that they might be candidate outcome measures for a future RCT”.

4) Unpublished data from the pilot RCT showed that the length of respiratory support is longer in patients who need intubation compared to those who do not. Therefore, the adverse impact of intubation is likely reflected in longer duration of respiratory support anyway.

5) The vast majority of centres participating in FIRST-ABC trial have just completed participation in the SANDWICH trial,² which provided protocolised management of weaning/extubation in invasively ventilated patients, which would have allowed for standardisation of practice across sites.

6) We agree that a large number of events may occur after intubation which affect the primary outcome, but we do not expect these to affect CPAP and HFNC randomised patients differently. We have added some of this context to the 'Outcome measures' section.

Another potential problem is the heterogeneity of PICU patients, age groups and comorbidities. With regards to analysis the binary division of age might not be sufficient, as the range from 13 months to 16 years of thoracic growth and therefore the subsequent FRC is very wide, and will differ greatly within the group of patient >12 months. It will greatly vary with onset of puberty as well. Age could be taken into analysis as an independent variable, when sub analyzing the outcome in the older group (>12 mo). Onset of puberty should have been taken into consideration. Apart from neuromuscular comorbidity, chronic lung disease should have been taken into account separately (not with the "other" group)

We anticipate that a high proportion (>50%) of patients will be in the <12-month age group. Accordingly, the expected number of patients who have reached puberty will be small, limiting exploration of this factor. We have added a sentence to confirm that we will explore using age as a continuous variable if there are sufficient patient numbers in the older age groups. The subgroup analyses will be reported as per the pre-specified definitions. Note that chronic lung diseases is included within the 'other respiratory' group for clinical indication subgroup in step-up RCT.

The practical issue of a center's familiarity with a particular mode of NIV is significant. If they have been using one mode for several years, training for the other mode should have been provided for the nursing staff. Please explain the decision not to.

During the preparatory phase of the trial, we ascertained from sites their usual practices for NIV and HFNC usage. Bar two PICUs, all used both treatments in routine practice. The two PICUs that used NIV infrequently undertook a local staff education program on the unit prior to recruitment.

In the step down arm, there is a very long time to start the NIV post extubation. The success of NIV might vary according to how soon after extubation it was started, to prevent atelectasis. Will the clinicians decision be clarified in the CRF? If NIV was started prophylactically, and child randomized prior to extubation, or were they extubated to air/NC oxygen and monitored for deterioration, prior the NIV was started? The time after extubation the NIV was started might be significant and should be recoded, as well as the clinicians reasoning for starting (prophylactic vs rescue)

We are collecting data on the time of extubation, randomisation and starting treatment. It will be possible to classify the step-down support into prophylactic and rescue based on the data. This will be reported in the results.

With regards to the quality of life questionnaire after discharge, it was not explained in the introduction how does this relate to the choice of NIV? Could this please be added to intro.

The quality of life questionnaire is part of the health economic evaluation. The use of this data is described in the 'Integrated health economic evaluation' section.

With regards to consent, it is stated that in the case of consent not obtained, the data will still be used. This is unusual. It is clear that the delayed consent is necessary. However, if not obtained after 2 attempts (letter or phone call), the data should not be used. Otherwise the whole purpose of consenting for research studies is completely bypassed. Please explain

The consent model used in FIRST-ABC has been directly informed by the CONseNt methods in paediatric Emergency and urgent Care Trials (CONNECT) study guidance⁴ and conduct of several paediatric critical care trials conducted by our team, all of which have had extensive Patient and Public Involvement and engagement. The “opt-out element” of the postal consent procedures was included based on recent evidence resulting from two qualitative interview studies conducted in the paediatric critical care setting (please see O’Hara et al., 2018⁵ and Peters et al., 2019⁶ references now added to the manuscript). Across the two studies, a total of 46 (13 bereaved) parents of critically ill children were interviewed. All parents found research without prior consent acceptable and indicated that they would have consented to use of their child’s information in the respective trials under investigation (one was a trial of temperature thresholds and the other, fluid bolus therapy), had their child been enrolled using deferred consent. We have evidence from the CATCH trial¹⁰, FiSh⁷ and CONNECT⁴ to believe that an alternative approach, such as a telephone call or ‘opt-in’ approach via a letter home is not supported by parents and will lead to missing data, which poses a significant threat to the validity of research and safety of participants. A published evaluation of the CATCH consent procedures¹¹ concluded that “RCTs should balance the potential burden that seeking consent in difficult situations may cause against risk of bias by disproportionately excluding children who die or are transferred.” Specifically, the majority of parents supported, and found acceptable, contact via post at 4 weeks and then at 8 weeks after death of their child, with a personalised ‘opt-out’ declaration - as long as the ‘opt-out’ declaration was emphasised in bold and the letter personalised, ideally by a known member of the clinical team. As this approach is deemed acceptable by parents, these consent procedures (including the personalised ‘optout’ element emphasised in bold), have now been used (and refined) in a number of recently completed paediatric critical care trials conducted by members of our team (FiSh⁷, Oxy-PICU⁸, Fever⁹, FIRST-ABC Pilot study¹), and supported by Patient and Public Involvement (parent) members of the respective Trial Management Groups. We also highlight that the ‘opt-out’ element of the consent procedures is only considered as a last resort (i.e. there is a low mortality rate in paediatric critical care and we expect that for the vast majority of survivors, parents/legal guardians will be approached for consent in hospital) and this is emphasized in study procedures and site training. We have added additional supporting references to the manuscript.

Last point, is the starting level of flow for the HF group. Starting the HF for infants larger than 3.5 kg will give HF of 7 L/min, which gives approximately equivalent PEEP to CPAP of 7 cmH₂O (Wilkinson et al, “Pharyngeal pressure with high-flow nasal cannulae in premature infants”, Journal of perinatology, 2008). If, on the other hand, the child is smaller, i.e just above 2 – 2.5kg, this might make clinicians start the HF at 4-5 l/min, per this study protocol. This would make the HF group disadvantaged in comparison to the amount of PEEP it provides. Therefore, infants with the admission weight below 3.5 kg should be excluded from analysis, or lowest starting HF rate should be 7 l/min.

There is no well validated approach to assessing the degree of PEEP provided by various flow rates in infants and children – variability in PEEP may result from mouth opening as well as flow rate and airway anatomy. Studies from Milesi et al. show that the average PEEP in bronchiolitis infants (not premature infants, who are not our study population) is around 4-5 cm H₂O for a flow rate of around 2 L/kg/min.¹² In addition, the beneficial effects of HFNC may not come solely from the PEEP, rather dead space washout may play a significant role too.

References

1. Ramnarayan P, Lister P, Dominguez T, et al. FIRST-line support for Assistance in Breathing in Children (FIRST-ABC): a multicentre pilot randomised controlled trial of high-flow nasal cannula therapy versus continuous positive airway pressure in paediatric critical care. Crit Care 2018;22(1):144. doi: 10.1186/s13054-018-2080-3 [published Online First:

2018/06/06]

2. Blackwood B, Agus A, Boyle R, et al. Sedation AND Weaning In Children (SANDWICH): protocol for a cluster randomised stepped wedge trial. *BMJ Open* 2019;9(11):e031630. doi: 10.1136/bmjopen-2019-031630
3. Zwarenstein M, Treweek S, Gagnier JJ, et al. Improving the reporting of pragmatic trials: an extension of the CONSORT statement. *BMJ* 2008;337:a2390. doi: 10.1136/bmj.a2390
4. CONSeNt methods in paediatric Emergency and urgent Care Trials: University of Liverpool; [Available from: <https://www.liverpool.ac.uk/psychology-health-and-society/research/connect/> accessed 24 April 2019.
5. O'Hara CB, Canter RR, Mouncey PR, et al. A qualitative feasibility study to inform a randomised controlled trial of fluid bolus therapy in septic shock. *Arch Dis Child* 2018;103(1):28-32. doi: 10.1136/archdischild-2016-312515 [published Online First: 2017/08/30]
6. Peters MJ, Khan I, Woolfall K, et al. Different temperature thresholds for antipyretic intervention in critically ill children with fever due to infection: the FEVER feasibility RCT. *Health Technol Assess* 2019;23(5):1-148. doi: 10.3310/hta23050 [published Online First: 2019/02/23]
7. Inwald DP, Canter R, Woolfall K, et al. Restricted fluid bolus volume in early septic shock: results of the Fluids in Shock pilot trial. *Arch Dis Child* 2019;104(5):426-31. doi: 10.1136/archdischild-2018-314924
8. Peters MJ, Jones GAL, Wiley D, et al. Conservative versus liberal oxygenation targets in critically ill children: the randomised multiple-centre pilot Oxy-PICU trial. *Intensive Care Med* 2018;44(8):1240-48. doi: 10.1007/s00134-018-5232-7 [published Online First: 2018/06/06]
9. Peters MJ, Woolfall K, Khan I, et al. Permissive versus restrictive temperature thresholds in critically ill children with fever and infection: a multicentre randomized clinical pilot trial. *Critical Care* 2019;23(1):69. doi: 10.1186/s13054-019-2354-4
10. Gilbert RE, Mok Q, Dwan K, et al. Impregnated central venous catheters for prevention of bloodstream infection in children (the CATCH trial): a randomised controlled trial. *Lancet* 2016;387(10029):1732-42. doi: 10.1016/s0140-6736(16)00340-8 [published Online First: 2016/03/08]
11. Harron K, Woolfall K, Dwan K, et al. Deferred Consent for Randomized Controlled Trials in Emergency Care Settings. *Pediatrics* 2015;136(5):e1316-22. doi: 10.1542/peds.2015-0512 [published Online First: 2015/10/07]
12. Milesi C, Baleine J, Matecki S, et al. Is treatment with a high flow nasal cannula effective in acute viral bronchiolitis? A physiologic study. *Intensive Care Med* 2013;39(6):1088-94. doi: 10.1007/s00134-013-2879-y [published Online First: 2013/03/16]

VERSION 2 – REVIEW

REVIEWER	Fulya Kamit İstanbul Gaziosmanpasa Hospital, Yeniyuzyil University
REVIEW RETURNED	25-Jun-2020

GENERAL COMMENTS	Thanks to the authors for the responses and their hard work. I read the responses to all reviewers, that clarified the matter about this pragmatic study.
---

REVIEWER	Andrea Wolfler Istituto Giannina Gaslini - Italy
REVIEW RETURNED	24-Jun-2020

GENERAL COMMENTS	All the issues have been addressed, thank you.
REVIEWER	Reinout A. Bem Pediatric Intensive Care Unit, Emma Children's Hospital, Amsterdam UMC, location AMC, Amsterdam, the Netherlands
REVIEW RETURNED	18-Jun-2020
GENERAL COMMENTS	Thank you for the opportunity to review the revised version of this study protocol manuscript. I believe the authors have revised the protocol description according to the suggestions of the five reviewers. My main concern over the choice of the primary outcome still stands (and with me reviewer 5 has this concern as well), but I understand the rationale and limitations as discussed by the authors. Given the importance of parental opinion in the choice of the primary outcome, it would be informative to actually study parental distress and anxiety in parents from children during support by non-invasive modalities versus invasive modalities (e.g. Klingenberg C et al 2014 studied HFNC versus CPAP), as I have different clinical experiences with parents toward this point. As such, I would like to stimulate an active discussion and in-depth (secondary) analysis on this issue by the authors once the study results and final documentation are obtained. I believe this study is important to the pediatric critical care field, and I would like to wish the trial investigators a successful trial.
REVIEWER	Dr Sanja Zivanovic University of Oxford, UK
REVIEW RETURNED	21-Jun-2020
GENERAL COMMENTS	The authors have explained well the reviewer's questions regarding the choice of primary outcome. They have further explained the plan for statistical analysis, particularly ways to overcome the variations in age groups that might affect the primary outcome. The consent procedure has been sufficiently elaborated on.